**Data Availability Statement:** All relevant data are within the paper.

**Funding:** This project is funded by the Department of Comparative Medicine at Stanford University.

# Effectiveness of two extended-release buprenorphine formulations during postoperative period in neonatal rats

**Mingyun Zhang**[1]*, **Eden Alamaw**[1], **Katechan Jampachaisri**[2], **Monika Huss**[1], **Cholawat Pacharinsak**[1]

**1** Department of Comparative Medicine, Stanford University School of Medicine, Stanford, California, United States of America, **2** Department of Mathematics, Naresuan University, Phitsanulok, Thailand

* mynzhang@stanford.edu

## Abstract

Information on the effectiveness of a new long-lasting buprenorphine formulation, extended-release buprenorphine, in the neonatal rat is very limited. This study compares whether a high dose of extended-release buprenorphine (XR-Hi) attenuates thermal hypersensitivity for a longer period than a low dose of extended-release buprenorphine (XR-Lo) in a neonatal rat incisional pain model. Two experiments were performed. Experiment one: Male and female postnatal day-5 rat pups (n = 38) were randomly assigned to 1 of 4 treatment groups and received a subcutaneous administration of one of the following: 1) 0.9%NaCl (Saline), 0.1 mL; 2) sustained release buprenorphine (Bup-SR), 1 mg/kg; 3) XR-Lo, 0.65 mg/kg; and 4) XR-Hi, 1.3 mg/kg. Pups were anesthetized with sevoflurane in 100% $O_2$ and a 5 mm long skin incision was made over the left lateral thigh and underlying muscle dissected. The skin was closed with surgical tissue glue. Thermal hypersensitivity testing (using a laser diode) and clinical observations were conducted 1 hour (h) prior to surgery and subsequently after 1, 4, 8, 24, 48, 72 h of treatment. Experiment two: The plasma buprenorphine concentration level was evaluated at 1, 4, 8, 24, 48, 72 h on five-day-old rat pups. Plasma buprenorphine concentration for all treatment groups remained above the clinically effective concentration of 1 ng/mL for at least 4 h in the Bup-SR group, 8 h in XR-Lo and 24 h in XR-Hi group with no abnormal clinical observations. This study demonstrates that XR-Hi did not attenuate postoperative thermal hypersensitivity for a longer period than XR-Lo in 5-day-old rats; XR-Hi attenuated postoperative thermal hypersensitivity for up to 4 h while Bup-SR and XR-Lo for at least 8 h in this model.

## Introduction

Postoperative analgesia is a critical component of laboratory animal medicine as untreated pain causes distress [1] and can impact research results [2]. Neonatal rodents are used for variety of survival surgical procedures, including stereotaxic [3–5], spinal [6, 7], and thoracic [8, 9] surgeries which are known to evoke pain responses. Untreated pain in neonates can lead to

The funder had no role in study design, data collection and analysis, decision to publish, or preparation of the manuscript.

**Competing interests:** The authors have declared that no competing interests exist.

long-term behavioral changes [10, 11], such as altered development of pain perception, neuro-developmental functioning, and social-emotional functioning [12]. Providing analgesia to neonates is challenging as there are significant physiological differences in neonates from adults. They have immature vital organs which can lead to rapid changes in their renal and hepatic systems [13]. This results in differing pharmacodynamics and an increased incidence of adverse effects from analgesic drugs, leading to urinary retention, sedation, and respiratory depression [14–17]. Because of this, additional consideration is needed prior to the use of analgesic drugs in neonates.

Opioid analgesics, including buprenorphine HCL (Bup-HCL, Hospira, Lake Forest, IL) and sustained-released buprenorphine (Bup-SR, Zoopharm, Fort Collins, CO), are the most commonly used analgesic drugs for managing post-operative pain in laboratory rodents [18]. In neonatal rats, a single dose of Bup-HCL (0.025 or 0.05 mg/kg) was found to effectively attenuate post-operative thermal hypersensitivity for 4 h [19], while a single dose Bup-SR (either 0.5 or 1 mg/kg) was found to effectively attenuate hypersensitivity for 8 h [20]. Recently, a new FDA-indexed, lipid-bound extended-release buprenorphine (XR-Hi, Ethiqa®, Fidelis Animal Health, North Brunswick, NJ) was introduced as an alternative buprenorphine analgesic for rodents. Bup-SR is a polymeric formulation that contains a water-insoluble, biodegradable lipid-bound and suspended in medium-chain fatty acid triglyceride (MCT) oil that is degraded over time with lipase and esterase activity [21–25]. In a previous study, our group found that a single dose of XR-Lo 0.65 mg/kg or XR-Hi 1.3 mg/kg, effectively attenuated post-operative mechanical hypersensitivity for 2 days in adult rats. However, the safety and efficacy of XR-Hi in neonatal rats is currently unknown.

In this study, we investigate the effectiveness of XR-Lo and high dose of extended-release buprenorphine (XR-Hi). We hypothesized that a high dose (1.3 mg/kg) of XR-Hi would attenuate post-operative thermal hypersensitivity longer than a low dose (0.65 mg/kg) of extended-release buprenorphine (XR-Lo, Fidelis Animal Health, North Brunswick, NJ) in a neonatal rat incisional pain model.

## Materials and methods

### Animals

One-day-old male and female Sprague Dawley rat pups [(Crl: CD (SD) IGS, (n = 94), Charles River Laboratories, Hollister, CA], housed in litters of eight to ten with the dam, arrived at the facility on day 0. The rats were free of rat coronavirus, rat Theiler virus, Kilham rat virus, rat parvovirus, Toolan H1 virus, rat minute virus, lymphocytic choriomeningitis virus, murine adenovirus type 1 and 2, reovirus type 3, Sendai virus, pneumonia virus of mice, *Mycoplasma pulmonis*, mites, lice, and pinworms. All animals were housed in static microisolator cages under a 12:12 light:dark cycle (lights on at 7:00 am). Rat dams were provided with Teklad Global 18% Protein Rodent Diet 2018 (Harlan Laboratories, Madison, WI) and water filtered by reverse osmosis *ad libitum*. All experiments in this study were approved by the Stanford University Administration Panel for Laboratory Animal Care. All animals were handled and housed according to the *Guide for the Care and Use of Laboratory Animals* [26] in a facility accredited by the Association for the Assessment and Accreditation of Laboratory Animal Care, International. At the conclusion of the study, rat pups and dams were euthanized with carbon dioxide asphyxiation followed by decapitation as a secondary method of euthanasia.

### Experimental groups

Five-day-old pups of either sex was randomly assigned into one of four treatment groups (n = 8–10 per group) and observers were blinded to the treatment group assignment.

Treatment groups consisted of: 1) Saline (Saline; 0.9% NaCl, Hospira, Lake Forest, IL) – 0.1 ml subcutaneous (SC) once ($n = 8$), 2) Bup-SR (1 mg/mL)– 1mg/kg SC once ($n = 10$), 3) XR-Lo (1.3 mg/mL) – 0.65 mg/kg SC once ($n = 10$), 4) XR-Hi 1.3 mg/kg SC once ($n = 10$). Doses of buprenorphine chosen were based on previous studies: 1) Bup-SR at 1 mg/kg was based on Blaney et al. [20]; 2) XR-Hi at 0.65 and 1.3 mg/kg was based on doses in adult Sprague Dawley rats by Levinson et al. [27] and Alamaw et al. [28] (note that doses of XR-Hi in rat pups are not known).

## Anesthesia and surgical model

Anesthesia was induced with sevoflurane (5–8%) in 100% $O_2$ (0.5–1 L/minute) via nose cone until a surgical plane of anesthesia was reached (absence of paw withdrawal reflex). Anesthesia was maintained with 4–7% sevoflurane in 100% $O_2$ (0.5 L/minute) via nose cone. The pups were placed on a warm water circulating blanket set to 38°C (Stryker T/ Pump, Portage, Michigan) for the duration of the anesthetic procedure. All drugs (Bup-SR, XR-Lo, and XR-Hi) were administered SC at the left shoulder. All rat pups were administered supplemental fluid (0.9% NaCl, 5 ml/kg, SC) at the right shoulder. XR-Lo and XR-Hi were administered with 29-gauge 0.3 ml insulin syringes (UltiMed, Inc., Excelsior, MN) while Bup-SR was administered with 22-gauge 1 ml Leur Lock syringe (Becton, Dickinson and Company, Franklin Lakes, NJ). All injection sites were pinched post-injection for 5 seconds (sec) to prevent leakage. All drug administrations were performed by the same personnel, who did not participate in subsequent testing or pathological evaluations, to ensure the tester is blinded to the treatment groups. The surgical method was adapted from Blaney et al. [20] and was performed aseptically as required by the Stanford IACUC Guidelines for Rodent Survival Surgery. Briefly, the pups were positioned in right (contralateral) recumbency, and the left (ipsilateral) thigh was aseptically prepared with three alternating betadine solution swab sticks (10% povidone-odine, Dynare Corporation, Orangeburg, NY) and 70% Isopropyl Alcohol USP (Henry Schein, Melville, NY). After the animal reached a surgical plane of anesthesia (as determined by a lack of withdrawal response to toe pinch), a 0.5-cm longitudinal skin incision was made on the ipsilateral thigh, 0.5-cm above the hock, using #15 blade (VetOne®, Boise, ID), and the superficial biceps femoris muscle was incised 0.3-cm longitudinally without disturbing the muscle attachment. The incision was closed with sterile surgical tissue glue (Covetrus, Dublin, OH). After surgery, antibiotic ointment (triple antibiotic ointment with bacitracin zinc, neomycin sulfate, and polymyxin B sulfate, Johnson & Johnson Consumer Inc., Skillan, NJ) was applied to the surgical area. Pups were then recovered in a warm recovery cage and placed back with the dam once fully recovered.

## Thermal hypersensitivity

Thermal hypersensitivity testing was performed by the same person, who is adequately trained in laser stimulation, throughout the experiment. Prior to the laser stimulation, skin temperatures of both the left (ipsilateral) and right (contralateral) thigh of each pup was measured with an infrared thermometer (Extech instruments, Nashua, NH). The skin temperature of both ipsilateral and contralateral side was 33.8 ± 0.1°C. Baseline thermal hypersensitivity of 5-day-old pups was determined at -1 h with laser stimulation adapted from Blaney et al. [20]. During stimulation and testing, two to three pups were evaluated at a time by maintaining them on a warm water circulating blanket (set at 38 °C) (Stryker T/Pump, Portage, Michigan, USA). Then, pups were tested with an infrared diode laser stimulator (LASMED, Mountain View, CA, USA, 490 mA, 5 mm diameter, 3.5 inch from skin) with a cutoff time of 19-sec to prevent skin burn. Thermal hypersensitivity was measured in sec as the time from when the laser was

started until a purposeful paw movement away from the laser. The right and left thighs were tested twice, at three-minute intervals, and these two measurements were averaged. Thermal hypersensitivity was defined as a significant decrease in withdrawal latency following focal thermal stimuli.

## Clinical observations and gross pathology

For the duration of the study, all pups were: 1) weighed daily at -1, 24, 48, 72 h(s); 2) observed for abnormal behaviors (i.e., general activity, mobility, maternal acceptance, and milk spots); 3) monitored for abnormalities at the injection site and incisional site. Gross pathology was performed at the end of the experiment.

## Plasma collection

In a separate experiment, pups (n = 56) were assigned to the same treatment groups as the surgery experiment. Pups were anesthetized with 5–8% sevoflurane and injected with either Saline 0.1 ml SC ($n$ = 2); Bup-SR 1 mg/kg SC ($n$ = 18); XR-Lo 0.65 mg/kg SC ($n$ = 18), or XR-Hi 1.3 mg/kg SC ($n$ = 18). All pups then recovered in a warm cage and were placed back with the dam once recovered. For blood collection, pups were deeply anesthetized with 5–8% sevoflurane in 100% $O_2$. After a surgical plane of anesthesia was confirmed by toe pinch, whole blood was collected by cardiac puncture using heparinized 1 mL tuberculin syringes and 25G needles. Blood was collected at 1, 4, 8, 24, 48, and 72 h(s) post- drug administration. For Saline group, blood collection was performed only at 1 h after injection. At the completion of blood collection, euthanasia was confirmed by cervical dislocation. Whole blood was collected (50 μL and up) in lithium heparinized microtainers and spun in a microcentrifuge at 2,500 rpm for 20 minutes. The plasma was collected into cryogenic tubes and stored in -80˚C prior to shipment for analysis.

## Plasma concentration analysis

Plasma samples were shipped on dry ice through overnight delivery to the McWhorter School of Pharmacy Pharmaceutical Sciences Research Institute (Samford University, Birmingham, AL) to analyze the plasma concentration of buprenorphine. The plasma concentrations of buprenorphine are obtained through liquid chromatography-tandem mass spectrometry (HPLC MS/MS). Buprenorphine standard spiking solutions were prepared by mixing deionized water and acetonitrile in 50:50 ratio to allow concentrations in plasma to range from 0.2– 200 ng/mL. The buprenorphine plasma samples and standards (100 μL) were fortified with internal standard (50 ng/mL terfenadine). To precipitate the plasma proteins, 1 mL acetonitrile was added. The mixture was then vortexed and centrifuged followed by organic layer transferred to clean test tube and evaporated under nitrogen in a water bath at 50˚C until dry. The samples were then reconstituted in dilution solvent and analyzed by HPLC MS/MS. Blank control plasma was used to prepare matrix matched standards and QC sample. The lower limit of detection for this assay was 0.1 ng/ml. Due to the small volume of whole blood collected from pups, not all samples could be analyzed for buprenorphine concentration. Each sample analyzed had a minimum volume of 50 μL. The sample size for most timepoints analyzed was $n$ = 3 but was $n$ = 2 in XR-Lo 48 h, Bup-SR 1 h, Bup-SR 24 h, and $n$ = 1 in XR-Lo 72 h and Bup-SR 72 h.

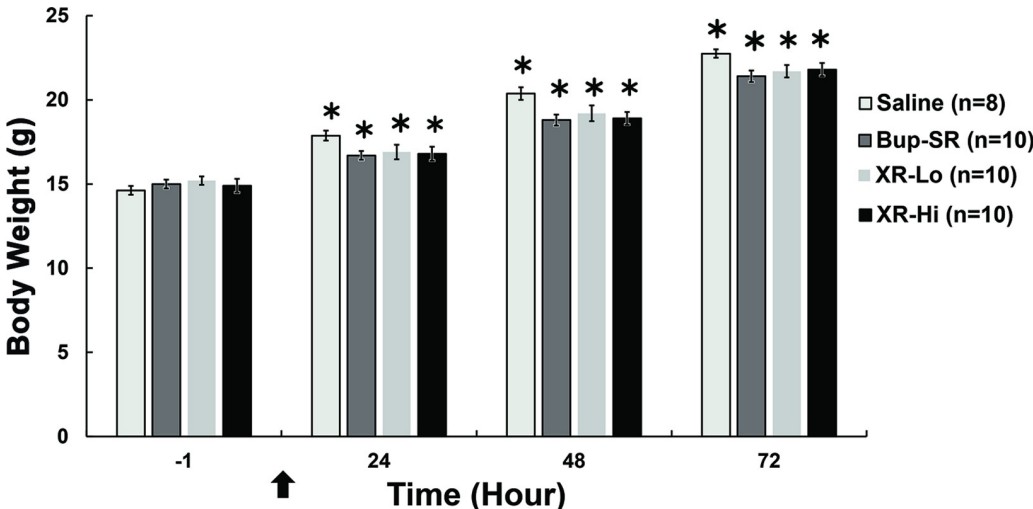

**Fig 1. Body weights of rat pups throughout the course of the study.** Mean body weight ± SEM. Arrow indicates the time of surgery. *Value significantly ($P<0.05$) increased compared to -1 h (baseline).

## Statistical analysis

To assess significances of differences in skin temperature, thermal hypersensitivity, and body weight, over time, an F test in ANOVA with repeated measures, followed by Bonferroni correction to examine differences within groups over time and between groups at the same time point. Data was also tested for normality using R software (R Core Team, 2021). The data for plasma concentration was not statistically analyzed. Data is presented as mean ± standard error of the mean (SEM). A $p$-value of less than 0.05 was considered significant.

## Results

### Body weight

There were no differences between sexes for weight, therefore data was combined. The body weight statistically increased in all treatment groups from -1 h (Saline = 14.6 ± 0.3 g; Bup-SR = 15 ± 0.3 g; XR-Lo = 15.2 ± 0.3 g; XR-Hi = 14.9 ± 0.4 g) to 72 h (Saline = 22.8 ± 0.3 g; Bup-SR = 21.4 ± 0.3 g; XR-Lo = 21.7 ± 0.4 g; XR-Hi = 21.8 ± 0.4 g). Baseline (-1 h) body weight of rat pups in Saline (14.6 ± 0.3 g), Bup-SR (15 ± 0.3 g), XR-Lo (15.2 ± 0.3 g) and XR-Hi (14.9 ± 0.4 g) groups was not significantly different (Fig 1). The weight of the Saline group at 24 h (17.9 ± 0.3 g) was significantly higher than that of the Bup-SR group (16.7 ± 0.3 g) and the XR-Hi group (16.8 ± 0.4 g) but was not significantly different between the four treatment groups at any other time point.

### Thermal hypersensitivity

There was no thermal hypersensitivity difference between sexes; therefore, data was combined. Thermal hypersensitivity did not differ at -1 h (before surgery, baseline) between groups in the ipsilateral or contralateral thigh (Fig 2).

   **Comparison within a drug group to baseline value (-1 h).** *Ipsilateral thigh.* Rat pups in the Saline group had significantly decreased (more sensitive) thermal latency at 1 (5.3 ± 0.4 sec), 4 (7.1 ± 0.7 sec), and 8 (6.9 ± 0.8 sec) h compared to –1 h (16.8 ± 1.4 sec; baseline). No significant differences were observed in the Bup-SR group at any time point compared to -1 h

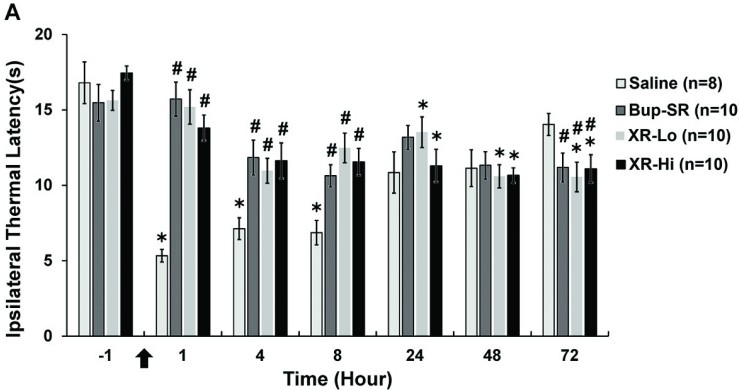

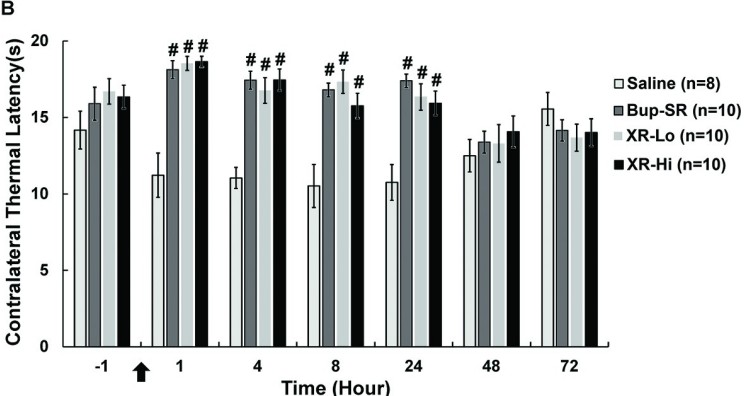

**Fig 2. Thermal hypersensitivity results.** Thermal hypersensitivity of the (**A**) ipsilateral thigh measured in sec; mean ± SEM; * = Values significantly (P<0.05) different from that of -1 h (baseline) for the same treatment group. # = Values significantly (P<0.05) different compared to the Saline group at the same time point. (**B**) Contralateral thigh measured in sec; mean ± SEM. * = Values significantly (P<0.05) different from that of -1 h (baseline) for the same treatment group. # = Values significantly (P<0.05) different compared to the Saline group at the same time point. Arrow indicates the time of surgery.

(15.5 ± 1.2 sec, baseline) in the ipsilateral thigh. Compared to its baseline values at -1 h (15.6 ± 0.7 sec), XR-Lo's thermal latency significantly decreased (more sensitive) at 48 (10.6 ± 0.8 sec), and 72 (10.6 ± 1.0 sec) h. Compared to its baseline values -1 h (17.4 ± 0.5 sec), XR-Hi's thermal latency significantly decreased (more sensitive) at 8 (11.6 ± 0.9 sec), 24 (11.3 ± 1.1 sec), 48 (10.7 ± 0.5 sec), 72 (11.1 ± 0.9 sec) h.

*Contralateral thigh.* No significant differences in thermal latency were observed between any group at any time points from their baseline measurements (Saline = 14.2 ± 1.2 sec; Bup-SR = 15.9 ± 1.1 sec; XR-Lo = 16.7 ± 0.8 sec; XR-Hi = 16.3 ± 0.8 sec).

**Comparison between drug groups at the same time point.** *Ipsilateral thigh.* Compared to the Saline group, the thermal latency for the Bup-SR, XR-Lo and XR-Hi groups were significantly increased (less sensitive) at 1 (Saline = 5.3 ± 0.4 sec; Bup-SR = 15.7 ± 1.1 sec; XR-Lo = 15.2 ± 1.1 sec; XR-Hi = 13.8 ± 0.9 sec), 4 (Saline = 7.1 ± 0.7 sec; Bup-SR = 11.8 ± 1.2 sec; XR-Lo = 11.0 ± 0.8 sec; XR-Hi = 11.6 ± 1.2 sec), 8 (Saline = 6.9 ± 0.8 sec; Bup-SR = 10.6 ± 0.7 sec; XR-Lo = 12.5 ± 1.0 sec; XR-Hi = 11.6 ± 0.9 sec), and decreased (more sensitive) at 72 (Saline = 14.0 ± 0.7 sec; Bup-SR = 11.2 ± 1.0 sec; XR-Lo = 10.6 ± 1.0 sec; XR-Hi = 11.1 ± 1.0 sec) h. The thermal latency for the Bup-SR, XR-Lo and XR-Hi groups was not significantly different between groups at any time point throughout the study.

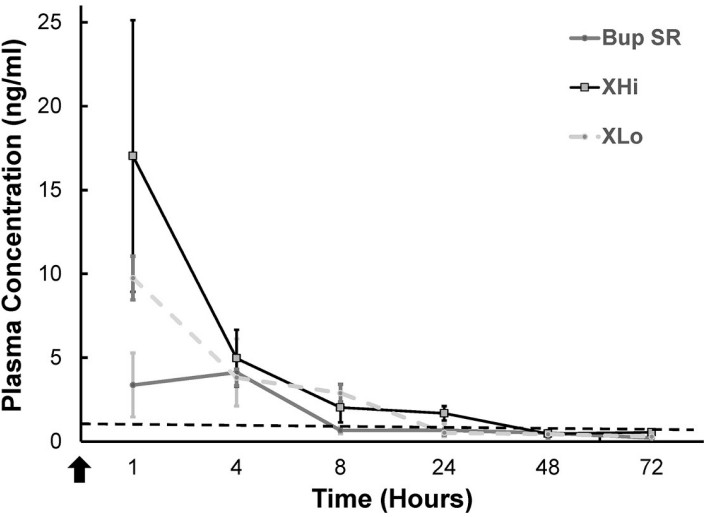

**Fig 3. Plasma concentration results of rat pups.** Plasma concentration (ng/mL, mean ± SEM) of Bup-SR, XR-Lo, and XR-Hi in treated rat pups (*n* = the number pups sampled at each time point). Samples were analyzed at 1, 4, 8, 24, 48, 72 h(s) after administration. *n* = 3 for all groups except *n* = 2 in XR-Lo at 48 h, Bup-SR at 1 h and Bup-SR at 24 h and *n* = 1 for XR-Lo at 72 h and Bup-SR at 72 h due to insufficient sample collected. Dotted line indicates clinically effective plasma concentration (1.0 ng/mL). Arrow indicates the time of drug administration.

*Contralateral thigh*. Compared to the Saline group, the thermal latency for the Bup-SR, XR-Lo and XR-Hi group was significantly increased (less sensitive) at 1 (Saline = 11.2 ± 1.4 sec; Bup-SR = 18.1 ± 0.6 sec; XR-Lo = 18.5 ± 0.5 sec; XR-Hi = 18.6 ± 0.4 sec), 4 (Saline = 11.0 ± 0.7 sec; Bup-SR = 17.4 ± 0.6 sec; XR-Lo = 16.8 ± 0.8 sec; XR-Hi = 17.5 ± 0.7 sec), 8 (Saline = 10.5 ± 1.4 sec; Bup-SR = 16.8 ± 0.4 sec; XR-Lo = 17.3 ± 0.8 sec; XR-Hi = 15.8 ± 0.8 sec) and 24 (Saline = 10.7 ± 1.2 sec; Bup-SR = 17.4 ± 0.4 sec; XR-Lo = 13.3 ± 1.2 sec; XR-Hi = 15.9 ± 0.8 sec) h. The thermal latency for the Bup-SR, XR-Lo and XR-Hi groups was not significantly different between groups at any time point throughout the study.

## Plasma drug concentration analysis

Buprenorphine plasma concentration remained above 1 ng/mL at 1 and 4 h(s) after drug administration for all three treatment groups (1 h: Bup-SR = 3.375 ± 1.905 ng/mL, XR-Lo = 9.8 ± 1.3 ng/mL, XR-Hi = 17.0 ± 8.1 ng/mL; 4 h: Bup-SR = 4.1 ± 2.0 ng/mL, XR-Lo = 3.8 ± 0.5 ng/mL, XR-Hi = 5.0 ± 1.7 ng/mL) (Fig 3). The plasma concentration decreased to below 1 ng/mL in Bup-SR group at 8 h, XR-Lo group at 24 h, and the XR-Hi group at 48 h post drug administration (Bup-SR = 8 h: 0.7 ± 0.2 ng/mL; XR-Lo = 24 h: 0.5 ± 0.1 ng/mL; XR-Hi = 48 h: 0.5 ± 0.2 ng/mL). Saline injected pups were used as negative controls at 1 h (0 ng/mL).

## Clinical observation and gross pathology

Decreased overall activity was observed for the first h after surgery in the Bup-SR (40%; 4/10 pups), XR-Lo (70%; 7/10 pups), and XR-Hi (70%; 7/10 pups) groups. Gross pathologic examination was performed at the end of the study at 72 h. At the drug injection sites (left shoulder), mass lesions characterized by firm ellipsoid nodules in the subcutis (0.4–1 cm) were observed in rat pups in the Bup-SR (40%; 4/10 pups), XR-Lo (40%; 4/10 pups), and XR-Hi (50%; 5/10 pups) treatment groups. No lesions were observed in the Saline group.

## Discussion

The aim of this study was to evaluate if XR-Hi attenuates post-operative thermal hypersensitivity longer than XR-Lo in an incisional pain model in 5-day-old neonatal rats. This study demonstrates that XR-Hi did not attenuate postoperative thermal hypersensitivity longer than XR-Lo in 5-day-old rats. In short, a single dose of XR-Lo (0.65 mg/kg SC) attenuated thermal hypersensitivity for up to 8 h and XR-Hi (1.3 mg/kg SC) attenuated thermal hypersensitivity for 4 h in a modified incisional pain model of neonatal rats. This is the first study evaluating the efficacy of extended-release buprenorphine (XR-Lo and XR-Hi) in a neonatal rat model of incisional pain. As a result, we recommend XR-Lo at a dose of 0.65 mg/kg for analgesic management of an incisional pain model in neonatal rodents.

We adopted a modified incisional pain model because our lab has extensive experience with evaluating analgesic efficacy using this model representing a minor pain surgical procedure in rodents. Post-operative thermal hypersensitivity has been observed for up to four days in adult rats when using the plantar incisional pain model [29–32]. However, due to the small hind paw size of neonatal rats, we modified the incisional pain model and performed a skin incision with dissection of the underlying muscle on the left thigh to increase the area for post-operative hypersensitivity testing. Previous studies by our lab have indicated that the thigh incisional pain model induces a post-operative thermal hypersensitivity response for 4 to 8 h in neonatal rats [19, 20]. Previous work indicated that a diode laser provides sufficient stimuli to cause a withdrawal reflex to evaluate thermal hypersensitivity in adult rats [33]. Using this technique, we found that post-operative thermal hypersensitivity lasted for 8 h in the Saline group, indicating that this thigh incisional model produced thermal hypersensitivity for at least 8 h in rat pups. The contralateral thigh was also tested to serve as an uninjured control during testing. In this current study, differences in thermal hypersensitivity testing on the contralateral thigh found in the Bup-SR, XR-Lo, and XR-Hi groups as compared to Saline group, at 1, 4, 8, and 24 h are likely due to sedation from the treatment drugs. Although sedation was not clinically evident in this study, it has been previously noted in the literature when administering Bup-SR or XR-Hi to rats [28, 29, 34].

Sustained release formulations of buprenorphine offer refinement to laboratory animal analgesia practices. Bup-SR has been frequently used to manage post-operative pain as it requires less frequent dosing and provides a sustained duration of analgesia [20, 25]. Bup-SR was found to attenuate post-surgical thermal hypersensitivity (with a laser diode test) in neonatal rats for at least 8 h using the thigh incisional pain model [20]. In this current study, we confirmed these results. Recently, a new long-lasting buprenorphine formulation (XR-Hi) that is FDA-indexed and cGMP-compliant, became commercially available. Although both Bup-SR and XR-Hi have an identical active ingredient, buprenorphine, differences in their formulations and release technology may affect the drug's pharmacokinetics. Therefore, we decided to evaluate XR-Hi as compared to Bup-SR in neonatal rodents. Our group has found that XR-Hi effectively attenuates post-operative pain for 48 h in adult mice [25] and adult rats [28] using the plantar incisional model. Such a difference between Bup-SR and XR-Hi in the duration of attenuating pain may be attributed to the technological differences between the two drugs. Bup-SR dissolves liquid polymer, with buprenorphine encapsulated inside, in a biocompatible organic solvent [25]. After injection, the polymer undergoes biodegradation, hydrolysis, and drug diffusion, resulting in the release of buprenorphine [24]. XR-Hi is contained in a lipid capsule, suspended in medium train fatty acid triglyceride. Unlike Bup-SR, buprenorphine in XR-Hi is released due to lipase and esterase activity [22]. This difference between Bup-SR and XR-Hi regarding how buprenorphine is released is one of the reasons for the different dosing adopted in this paper.

Due to developmental differences between neonates and adults, neonatal rodent pups are known to be more sensitive to anesthetics and analgesics. Analgesic dosages for neonatal pups should be adjusted to reflect the appropriate clinically effective dose for their age. Therefore, in this current study, to evaluate the safety, dosing, and efficacy of this new analgesic drug in neonatal pups, we also evaluated the low and high dose of XR-Hi which are efficacious for adult rats [28]. The XR-Lo group received the manufacturer's recommended dose for rats (0.65 mg/kg) and the XR-Hi group received two times (1.3 mg/kg) the recommended dose. We found that XR-Lo attenuated thermal hypersensitivity up to 8 h while XR-Hi only attenuated thermal hypersensitivity until 4 h.

In the present study, both the XR-Lo and XR-Hi groups achieved peak plasma concentration at 1 h (XR-Lo = 9.8 ± 1.3 ng/mL; XR-Hi = 17.0 ± 8.1 ng/mL) and the Bup-SR group achieved peak concentration at 4 h (4.1 ± 2.0 ng/mL). At 8 h, the plasma concentration of Bup-SR was 0.7 ± 0.2 ng/mL and XR-Lo was 2.9 ± 0.5 ng/mL. During hypersensitivity testing at the 8 h timepoint, we found that both Bup-SR and XR-Lo groups attenuated thermal hypersensitivity. At 0.7 ng/mL, the buprenorphine concentration of Bup-SR at 8 h is near the clinically effective plasma concentration (1 ng/mL) and previous work has indicated that a lower effective plasma concentration of opioids may provide sufficient analgesia for neonates [35]. Although the clinically effective plasma buprenorphine for adult rats is reported to be at least 1 ng/mL [36, 37], in this modified incisional pain model, we found the effective plasma buprenorphine for neonatal rat pups to be at least 0.7 ± 0.2 ng/mL. To our knowledge, this is the first study determining the effective plasma buprenorphine concentration for attenuation of thermal hypersensitivity in five-day-old Sprague Dawley rats.

In this study, we did not find a clear correlation between the plasma buprenorphine level and hypersensitivity attenuation observed at each timepoint. For example, at 8 h the XR-Hi group had plasma concentration of 2.0 ± 0.9 ng/mL and thermal hypersensitivity was also observed on the ipsilateral thigh for this treatment group. Thermal hypersensitivity was also observed in XR-Hi treatment groups on ipsilateral thigh at 8 (XR-Hi) and 24 (XR-Hi) h despite plasma buprenorphine levels measuring above 1 ng/mL for these time points. One possible explanation for this discrepancy is that buprenorphine can have a low receptor binding rate leading to an unequal correlation between the plasma concentration of buprenorphine and the degree of analgesia [38]. Other possible reasons are that high levels of plasma buprenorphine may result in nociceptin induced hyperalgesia [39, 40] and/or opioid induced hyperalgesia [41]. Nociception induced hyperalgesia may be the result of buprenorphine interacting with the opioid receptor-like (ORL-1) receptor [39, 42, 43]. Nociceptin is structurally similar to the endogenous opioid peptide, is the endogenous ligand for ORL-1 receptor, and is also known to block the antinociceptive effects induced through μ, δ, and κ opioid receptors [44]. Opioid induced hyperalgesia (OIH), is a paradox where the ongoing or increased administration of an opioid leads to an unexpected increased pain perception and sensitivity [45]. Neonates in particular are known to have increased sensitivity to the effect of opioids [46] and OIH has been described in human neonates [45] and rat neonates [47]. Additionally, hyperalgesia has been previously associated with administration of low or very high opiate dosage such as 10 times higher than recommended dose [41, 48, 49]. We found this level of buprenorphine in the plasma for the XR-Lo (9.8 ± 1.3 ng/mL) and XR-Hi (17.0 ± 8.1 ng/mL) group at 1 h. OIH is also known to have dosage-dependent effects: the higher the dose administered, the more hyperalgesia observed [50, 51]. The administration of a large dose of opioid can increase the frequency and onset time of OIH [52, 53]. We suspect the fast onset of OIH in XR-Hi group at 8 h post-operatively is due to the high dosage administered, and that at the lower dosage (XR-Lo group), OIH was not observed until 48 h after drug administration.

While generally safe, administration of opioid drugs has been reported to produce nausea [54, 55], constipation [56, 57], vomiting [54, 58], body weight change [59, 60], and impaired gastrointestinal motility [61] in a variety of species which can lead to changes in body weight. Body weight is commonly used as a measurement of the post-operative well-being of research animals and can be used to access adverse effects following drug administration [60, 62]. After surgery and drug administration, all pups were accepted by the dam as evidenced by the presence of a milk spot and increased weight gain. Additional clinical signs that may be associated with buprenorphine include respiratory depression [63], pica [64], cardiovascular depression [65], and sedation [66]. In this study, cardiovascular or respiratory depression and pica were not observed. However, at 1 h post-surgery, decreased general activity was observed in 40% of pups in the Bup-SR group, 70% of the pups in XR-Lo and XR-Hi groups, and 10% of pups in the Saline group. The decreased general activity in the Saline group at 1 h might be due to residual anesthetic effects. A small number of pups still had decreased activity at 4 h post-drug administration (10% of Bup-SR and 20% of XR-Lo). This observation indicates that buprenorphine was likely responsible for some sedation after administration.

Sustained release formulations of analgesic drugs have been reported to lead to skin irritation and injection site reactions to both rats and mice [67–69]. Although in a previous study by our lab evaluating XR-Hi in adult mice, we did not find any gross pathologic abnormalities or injection site reactions [25]. A study by our lab investigated the effect of XR-Hi in adult rats and found subcutaneous nodules at the site of administration of XR-Hi and Bup-SR [28]. In this current study, although it was not clinically evident, we also found subcutaneous nodules at the time of gross pathological examination in the Bup-SR, XR-Lo, and XR-Hi groups. We suspect that the nodules can be attributed to the polymer matrix that is used in sustained release formulations [69]. In addition, previous studies have shown prolonged hyperalgesia to be observed up to 5 to 12 days after administration of very high dose and repeated subcutaneous injection of opioids in adult rats [41, 70, 71].

A limitation of this study is that only thermal and not mechanical hypersensitivity was evaluated, even though mechanical hypersensitivity is known to be present in neonatal rats [72, 73]. Von Frey monofilament test is commonly used for mechanical hypersensitivity [74]. Despite von Frey monofilament had been used to test for pain

(-like) hypersensitivity in neonatal rats [75], previous work from our lab indicated that this test was impractical and had inconsistent results with neonatal rats [19]. During a pilot study, the Randall-Selitto analgesiometer (Ugo Basile, Comerio, Italy) was also performed. This method aims to quantify the mechanical hypersensitivity threshold using an electronic device that quantifies the force applied. This method is commonly used in adult [76, 77] and neonatal rats [78]. However, we found that it was difficult to perform this test on the thigh and tail of neonatal rats, to identify painful behaviors with pups restrained, and to obtain consistent results. Future studies should also look at the effect of lower doses of XR-Hi in alleviating pain in neonatal rats to avoid OIH.

From a practical standpoint, assuming a 15 g pup, the current cost of XR-Hi ($0.65 mg/kg) is US $0.94, or for XR-Hi (1.3 mg/kg) US $1.88, based on list price. For comparison, the current cost of Bup-SR (1 mg/kg) is US $0.35. The cost estimated excludes human labor and clinical supplies required for the injection. Although Bup-SR is less expensive compared to XR, some institutions may elect to use XR-Hi because of requirements to use of cGMP compliant and FDA approved drugs. Furthermore, some readers may prefer XR-Hi since it is slightly less viscous than Bup-SR and is easier to draw up.

In summary, we found that a single dose of XR-Lo was safe and effective at attenuating post-operative thermal hypersensitivity in 5-day-old rat pups for at least 8 h for this model. XR-Hi was effective at attenuating thermal hypersensitivity until 4 h but resulted in opioid

induced hypersensitivity, therefore, this dose is not recommended. Future studies should continue to evaluate the dosing, duration, and effectiveness of sustained release analgesics in neonatal rodents using a model that causes sustained hypersensitivity.

## Supporting information

**S1 Data.**
(XLSX)

## Acknowledgments

The authors thank Janis Atuk-Jones, Lisa Bandini, and Sonja Goedde for their technical and editing help with the article. We would like to thank the VSC Animal Diagnostic Laboratory, Department of Comparative Medicine, Stanford University, Benjamin Franco, Elias Godoy, Kaela Navarro, Alexandra Blaney, Marlon Pailano, Dr. David Yeomans, Dr. Michael Klukinov and Dr. Gregory Gorman at Samford University (plasma analysis) for their assistance with our investigation.

## Author Contributions

**Conceptualization:** Mingyun Zhang, Eden Alamaw, Monika Huss, Cholawat Pacharinsak.

**Formal analysis:** Katechan Jampachaisri.

**Investigation:** Mingyun Zhang, Eden Alamaw, Monika Huss, Cholawat Pacharinsak.

**Methodology:** Cholawat Pacharinsak.

**Project administration:** Monika Huss.

**Supervision:** Cholawat Pacharinsak.

**Visualization:** Mingyun Zhang.

**Writing – original draft:** Mingyun Zhang.

**Writing – review & editing:** Eden Alamaw, Monika Huss, Cholawat Pacharinsak.

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
