## [Decision Letter · Decision Letter 0]

11 Aug 2022

PONE-D-22-06559Extended-release buprenorphine effectively attenuates thermal hypersensitivity in an incisional model in neonatal rats (Rattus norvegicus)PLOS ONE

Dear Dr. Zhang,

Thank you for submitting your manuscript to PLOS ONE. After careful consideration, we feel that it has merit but does not fully meet PLOS ONE’s publication criteria as it currently stands. Therefore, we invite you to submit a revised version of the manuscript that addresses the points raised during the review process.

We look forward to receiving your revised manuscript.

Kind regards,

Sairah Hafeez Kamran, PhD

Academic Editor

PLOS ONE

Journal Requirements:

Reviewers' comments:

Reviewer's Responses to Questions

**Comments to the Author**

1. Is the manuscript technically sound, and do the data support the conclusions?

Reviewer #1: Partly

Reviewer #2: Yes

2. Has the statistical analysis been performed appropriately and rigorously? 

Reviewer #1: Yes

Reviewer #2: Yes

3. Have the authors made all data underlying the findings in their manuscript fully available?

Reviewer #1: Yes

Reviewer #2: Yes

4. Is the manuscript presented in an intelligible fashion and written in standard English?

Reviewer #1: No

Reviewer #2: Yes

5. Review Comments to the Author

Reviewer #1: Dear author.

If the authors consider the following points, the manuscript will become more valuable:

1. The title is inconsistent; if possible, rewrite it to make it more clear.

2. The abstract is vague and inconsistent; the goal, experiment protocol, and groups are unclear; it should be rewritten. Abbreviations such as XR-Hi, XR-Lo, and XR should be placed appropriately. Introduction: from lines 50-53, these are results, that must be put in the result section.

Add some lines for the differences between Bup-SR and Bup-XR and why the author chooses these doses.

3. Experimental groups: lines 84-85 the dose of the group is 0.65 mg/ kg low dose extended-release buprenorphine (XR-Lo; ), but write 1.3 mg/kg, please confirm which dose is correct. Additionally, this section needs rewriting to be consistent. Why does the author use these doses?

4. Anesthesia and surgical section: lines 93-95, Then all experimental treatments and supplemental fluid (0.9% NaCl, 5 ml/kg) were administered SC at the left and right shoulders, respectively. This section is not obvious, please rewrite this section.

5. Thermal Hypersensitivity Testing: in line 116, write 700 AM, what are the authors mean here? Furthermore, this section is vague.

6. In plasma collection: this part of the study why separated from the first part?

7. Plasma Concentration Analysis: abbreviation of DI to what refer?

8. Results: In line 214, the authors write there is a significant decrease in the thermal hypersensitivity in times 1,4, and 8 hrs, but in figure 2 A there is an increase in the thermal hypersensitivity, please give an explanation for this increment. Please, all results need checking.

9. Discussion:

a. The aim written in line 264 differs from that in the introduction and abstract. Confirm which one is correct.

b. Write whole word before abbreviation of Ethiqa-xr.

c. Many sentences are repeated frequently in the manuscript, so, revisions are required. See lines 273- 275 and the paragraph before these lines as an example.

d. The authors write the tested drug is safely and effectively attenuated the thermal hypersensitivity, please how measure these parameters?

e. Lines 285-287 are related to the materials and method section.

f. The word Saline should be used in uniform, because some times write with capital letter and other times with small letter.

g. From line 296 to 310 are related to the introduction section.

h. Lines 325- 337 are repeated and related to results section.

With best regards,

Reviewer #2: Report on manuscript

Extended-release buprenorphine effectively attenuates thermal hypersensitivity in an incisional model in neonatal rats (Rattus norvegicus)

Manuscript ID: PONE-D-22-06559

Journal: PLOS ONE

Article type: Original Article

Thank you for the opportunity to review this manuscript. Current manuscript by Zhang et al, reports the analgesic effect of extended-release (XR) buprenorphine on thermal hypersensitivity in neonatal rats. Overall, the article addresses an interesting question: effectiveness of analgesic techniques in neonatal rats to manage post-operative pain. The authors demonstrate that low dose of XR-buprenorphine attenuate post-operative thermal hypersensitivity for longer than a high dose XR-buprenorphine.

Abstract is written in a precise way.

Introduction part is well-structured making sufficient connection between literature survey and research question.

Methodology is feasible and provides sufficient details. Study design is appropriate.

Results and discussion portions are described in an appropriate fashion.

Major Concern:

However, I have some concerns. Authors have already published their research data on “buprenorphine”/ “sustained-released buprenorphine” as an analgesic drug in rats/neonatal rats/mice (Reference # 19, 20, 22, 24, 30, 31). Further, analgesic effect of “XR-buprenorphine” on mechanical hypersensitivity in rats, was investigated and published (Reference 29). Current study is the continuation of the same research strategy involving XR-buprenorphine in neonatal rat model. Limiting point of this study is that this work is lacking novelty. Most of the work is repetitive based on the parameters of the previous publications and data. Can authors justify the novelty of current work in a more elaborating way? Another limitation (as described in discussion) is mechanical hypersensitivity which is not evaluated. This should be considered to reach a conclusive argument about the analgesic effect of the drug. This work is interesting and after improvement this can be helpful to have an insight into the improved analgesic techniques, in lab animal models.

Minor concerns:

• Abbreviation should be spelled out in the first instance. For example: Abstract Line 16: “Bup SR”.

• Line 24: Re-phrasing. “Subsequently after 1,4,8,24,48, 72 hr of treatment or subsequently post-treatment of….”

• Line 403: For comparison, the current “cost” not “cause”.

6. PLOS authors have the option to publish the peer review history of their article (what does this mean?). If published, this will include your full peer review and any attached files.

Reviewer #1: **Yes: **Waleed K. Abdulsahib

Reviewer #2: No

---

## [Author Response · Author response to Decision Letter 0]

12 Sep 2022

Reviewer #1: Dear author.

If the authors consider the following points, the manuscript will become more valuable:

1. The title is inconsistent; if possible, rewrite it to make it more clear.

Response: The new title is renamed to make it clearer, “Effectiveness of Two Extended-Release Buprenorphine Formulations during Postoperative Period in Neonatal Rats”. 

2. The abstract is vague and inconsistent; the goal, experiment protocol, and groups are unclear; it should be rewritten. Abbreviations such as XR-Hi, XR-Lo, and XR should be placed appropriately. Introduction: from lines 50-53, these are results, that must be put in the result section. Add some lines for the differences between Bup-SR and Bup-XR and why the author chooses these doses.

Response: Abstract is rewritten to make the goal, experiment protocol and groups consistent. The abbreviation for extended-release buprenorphine has been modified to Bup-XR to be consistent with previous published studies done by our institution. The abbreviations are now listed when the drug was first mentioned. .

Line 58-61 “Our group found that a single dose of 0.65 or 1.3 mg/kg extended-release

buprenorphine (XR), effectively attenuated post-operative mechanical hypersensitivity for 2 days in adult rats (data not shown). However, the safety and efficacy of extended-release buprenorphine in neonatal rats is currently unknown” are results of previous study or known information. “In a previous study,” has been added to the sentence to clarify this.

Some lines for the differences between Bup-SR and Bup-XR were added in the Introduction, “Bup-SR is a polymeric formulation that contains a water-insoluble, biodegradable polymer encapsulating buprenorphine and a biocompatible organic solvent. Bup-XR is lipid-bound and suspended in medium chain fatty acid triglyceride (MCT) oil that is degraded overtime with lipase and esterase activity” 

“Doses of buprenorphine chosen were based on previous studies: 1) Bup-SR at 1 mg/kg was based on Blaney et al. (PMID 33534864) study; 2) Bup-XR at 0.65 and 1.3 mg/kg was based on doses in adult Sprague Dawley rats by Levinson et al (PMID 34183094) and Alamaw et al. (PMID 34903316) (note that doses of Bup XR in rat pups are not known).” This was also added into Materials and Methods section.

3. Experimental groups: lines 84-85 the dose of the group is 0.65 mg/ kg low dose extended-release buprenorphine (XR-Lo; ), but write 1.3 mg/kg, please confirm which dose is correct. Additionally, this section needs rewriting to be consistent. Why does the author use these doses?

Response: Lines 89-90, 1.3 mg/ml, a concentration (mg/ml) of the drug, was correct. This section was rewritten to be consistent. Doses used were explained in previous questions and added into “Materials and Methods”

4. Anesthesia and surgical section: lines 93-95, Then all experimental treatments and supplemental fluid (0.9% NaCl, 5 ml/kg) were administered SC at the left and right shoulders, respectively. This section is not obvious, please rewrite this section.

Response: This part was rewritten “All drugs (Bup-HCL, XR-Lo, and XR-Hi) were administered (SC) at the left shoulder. All rat pups were administered supplemental fluid (0.9% NaCl, 5 ml/kg, SC at the right shoulder).”

5. Thermal Hypersensitivity Testing: in line 116, write 700 AM, what are the authors mean here? Furthermore, this section is vague.

Response: We have revised accordingly

6. In plasma collection: this part of the study why separated from the first part?

Response: In plasma collection, this part of the study was separated from the first part because, to collect sufficient plasma (blood) volume for plasma buprenorphine concentration analysis in rat pups (approximately 15 grams), a terminal blood collection must be performed. Therefore, this part of the study was separated from the first part.

7. Plasma Concentration Analysis: abbreviation of DI to what refer?

Response: DI is “deionized” water. This is revised.

8. Results: In line 214, the authors write there is a significant decrease in the thermal hypersensitivity in times 1,4, and 8 hrs, but in figure 2 A there is an increase in the thermal hypersensitivity, please give an explanation for this increment. Please, all results need checking.

Response: Line 214 was revised to make it clearer, “there was a significant decrease in the thermal latency time at 1, 4, and 8 h”. The explanation is included in the discussion (line 218-220). In short, this incisional pain model decreased thermal latency (more painful and shorter response time) at 1, 4 and 8 h post-surgery in Saline (control) group which was similar to a study reported by Blaney et al. (PMID 33534864). 

9. Discussion:

a. The aim written in line 264 differs from that in the introduction and abstract. Confirm which one is correct.

Response: The aim written in line 264, introduction and abstract were revised. 

b. Write whole word before abbreviation of Ethiqa-xr.

Response: Ethiqa-xr was spelled out (revised)

c. Many sentences are repeated frequently in the manuscript, so, revisions are required. See lines 273- 275 and the paragraph before these lines as an example.

Response: Repeated sentences and lines 273-275 were revised. 

d. The authors write the tested drug is safely and effectively attenuated the thermal hypersensitivity, please how measure these parameters?

Response: The tested drugs were evaluated for safety and efficacy through the determination that: 1) Thermal latency times of treatment drug groups on the ipsilateral thigh was not different from baseline (D-1) values at 4 and 8 (except for XR-Hi) h; 2) thermal latency times of treatment drug groups were higher than those of the Saline group at 4 and 8 h; 3) treatment drugs did not alter latency times on non-injured (contralateral) thighs; 4) maternal acceptance and the presence of a milk spot was evident in all pups (maternal acceptance including the presence of a milk spot is key for pup survival) (PMID 33534864); 5) All pups gained weight throughout the study (weight has been used to measure well-being PMID 31896391).; 5) Gross necropsies (in internal organs such as liver, kidneys etc) did not find significant or abnormal changes.

e. Lines 285-287 are related to the materials and method section.

Response: Lines 285-287 were moved to Materials and Methods section. 

f. The word Saline should be used in uniform, because sometimes write with capital letter and other times with small letter.

Response: “Saline” was revised (uniformed). 

g. From line 296 to 310 are related to the introduction section.

Response: Although line 296-310 is related to introduction, it is also appropriate to be present in the discussion section. Therefore, the authors decided to keep this section for the discussion. 

h. Lines 325- 337 are repeated and related to results section.

Response: For lines 325-337, although this is similar to the result section, in the discussion these results are scientifically and analytically further summarized by addressing what new knowledge was found in the current study and what is previously known from other studies. 

Reviewer #2: Report on manuscript

However, I have some concerns. Authors have already published their research data on “buprenorphine”/ “sustained-released buprenorphine” as an analgesic drug in rats/neonatal rats/mice (Reference # 19, 20, 22, 24, 30, 31). Further, analgesic effect of “XR-buprenorphine” on mechanical hypersensitivity in rats, was investigated and published (Reference 29). Current study is the continuation of the same research strategy involving XR-buprenorphine in neonatal rat model. Limiting point of this study is that this work is lacking novelty. Most of the work is repetitive based on the parameters of the previous publications and data. Can authors justify the novelty of current work in a more elaborating way? Another limitation (as described in discussion) is mechanical hypersensitivity which is not evaluated. This should be considered to reach a conclusive argument about the analgesic effect of the drug. This work is interesting and after improvement this can be helpful to have an insight into the improved analgesic techniques, in lab animal models.

Response: Although this current study is the continuation of the same research strategy, the novelty of the current study includes: 1) this is the first study demonstrating a newly marketed rodent FDA-indexed (pharmaceutical grade) extended-release buprenorphine (XR) in rat pups. This current study firstly showed that thermal hypersensitivity was attenuated for 4 h in XR-Hi and 8 h in XR-Lo groups. There is limited knowledge regarding the use of any extended-release formulation of buprenorphine in neonatal rodents. Different formulations of drugs can affect dosing regimen and effectiveness of the drug (PMID 25787030; 26854975). Although Bup-SR is an effective analgesic in adult rodents or other species, an effectiveness of XR (a different extended-release formulation of buprenorphine) in neonatal rats is not known; 2) this current study is the first to show that XR-Lo is as effective as commonly used sustained-release buprenorphine, Bup-SR, in rat pups; 3) this current study showed that the new formulation, XR, did not cause any skin reactions as have been observed with the use of Bup-SR (PMID 23294889; 21439213), hyperactivity observed was also observed in adult rodents (PMID 24459403; 34179720) and did not affect maternal acceptance (the key for pups survival and not evaluated in adult rodent studies). Although the parameters seemed to be repetitive, in pups, reliable and short duration hypersensitivity testing modalities were still not known. In addition, testing duration was crucial in pups because pups should not be removed too long from the dam due to potential complications of hypothermia and stress for both the pups and dam which can affect maternal acceptance or potentially lead to cannibalism. Therefore, the testing method chosen was based on what our group and others have published. Moreover, different surgical pain models cause different hypersensitivity responses; therefore, to be able to compare results to previous studies, a similar surgical model was selected. We are among the first groups to perform buprenorphine extended release testing in rat pups. 

• Abbreviation should be spelled out in the first instance. For example: Abstract Line 16: “Bup SR”.

Response: Abbreviations were spelled out. 

• Line 24: Re-phrasing. “Subsequently after 1,4,8,24,48, 72 hr of treatment or subsequently post-treatment of….”

Response: Line 24, re-phrasing was revised “subsequently after 1,4,8,24,48, 72 h of treatment”

• Line 403: For comparison, the current “cost” not “cause”.

Response: Line 403, “cause” was revised to “cost”.

---

## [Decision Letter · Decision Letter 1]

26 Sep 2022

PONE-D-22-06559R1Effectiveness of Two Extended-Release Buprenorphine Formulations during Postoperative Period in Neonatal RatsPLOS ONE

Dear Dr. Zhang,

Thank you for submitting your manuscript to PLOS ONE. After careful consideration, we feel that it has merit but does not fully meet PLOS ONE’s publication criteria as it currently stands. Therefore, we invite you to submit a revised version of the manuscript that addresses the points raised during the review process.

ACADEMIC EDITOR: I will suggest to increase the clarity of discussion by incorporating scientific explanations of the research question and correct minor grammatical mistakes. The resolution of figures shall be increased to enhance clarity.

We look forward to receiving your revised manuscript.

Kind regards,

Sairah Hafeez Kamran, PhD

Academic Editor

PLOS ONE

Journal Requirements:

Reviewers' comments:

Reviewer's Responses to Questions

**Comments to the Author**

1. If the authors have adequately addressed your comments raised in a previous round of review and you feel that this manuscript is now acceptable for publication, you may indicate that here to bypass the “Comments to the Author” section, enter your conflict of interest statement in the “Confidential to Editor” section, and submit your "Accept" recommendation.

Reviewer #1: (No Response)

Reviewer #2: All comments have been addressed

2. Is the manuscript technically sound, and do the data support the conclusions?

Reviewer #1: Partly

Reviewer #2: Yes

3. Has the statistical analysis been performed appropriately and rigorously? 

Reviewer #1: I Don't Know

Reviewer #2: Yes

4. Have the authors made all data underlying the findings in their manuscript fully available?

Reviewer #1: Yes

Reviewer #2: Yes

5. Is the manuscript presented in an intelligible fashion and written in standard English?

Reviewer #1: No

Reviewer #2: Yes

6. Review Comments to the Author

Reviewer #1: Dear author,

Many sections of the manuscript are still vague and uninformative. The abstract is still vague and lacks a conclusion.

with regards,

Reviewer #2: Authors have made the required changes in the manuscript as suggested. Abstract is looking better now. I am satisfied with the explanation of authors regarding novelty of the research question. I will suggest to include this explanation in discussion to reveal its novelty and mentioning the limitations. I will also suggest to improve the resolution of the figures.

7. PLOS authors have the option to publish the peer review history of their article (what does this mean?). If published, this will include your full peer review and any attached files.

Reviewer #1: **Yes: **Waleed K. Abdulsahib

Reviewer #2: No

---

## [Author Response · Author response to Decision Letter 1]

30 Sep 2022

ACADEMIC EDITOR:

I will suggest to increase the clarity of discussion by incorporating scientific explanations of the research question and correct minor grammatical mistakes. The resolution of figures shall be increased to enhance clarity.

Response: We revised our Discussion section accordingly. The manuscript was proofread, and grammatical mistakes are corrected. The resolution of figures has been increased. 

Reviewers' comments:

Reviewer's Responses to Questions

Comments to the Author

1. If the authors have adequately addressed your comments raised in a previous round of review and you feel that this manuscript is now acceptable for publication, you may indicate that here to bypass the “Comments to the Author” section, enter your conflict of interest statement in the “Confidential to Editor” section, and submit your "Accept" recommendation.

Reviewer #1: (No Response)

Reviewer #2: All comments have been addressed

2. Is the manuscript technically sound, and do the data support the conclusions?

Reviewer #1: Partly

Reviewer #2: Yes

Response: We have revised accordingly.

3. Has the statistical analysis been performed appropriately and rigorously?

Reviewer #1: I Don't Know

Reviewer #2: Yes

Response: Our statistical analysis was performed by Dr. Katechan Jampachaisri, a statistician, from the Department of Mathematics, Naresuan University.

4. Have the authors made all data underlying the findings in their manuscript fully available?

Reviewer #1: Yes

Reviewer #2: Yes

5. Is the manuscript presented in an intelligible fashion and written in standard English?

Reviewer #1: No

Reviewer #2: Yes

Response: We revised the manuscript and typographical or grammatical errors were corrected. 

6. Review Comments to the Author

Reviewer #1: Dear author,

Many sections of the manuscript are still vague and uninformative. The abstract is still vague and lacks a conclusion.

with regards,

Response: We revised the Abstract and Discussion sections. 

Reviewer #2: Authors have made the required changes in the manuscript as suggested. Abstract is looking better now. I am satisfied with the explanation of authors regarding novelty of the research question. I will suggest to include this explanation in discussion to reveal its novelty and mentioning the limitations. I will also suggest to improve the resolution of the figures.

Response: We revised Discussion section accordingly. We also improved the resolution of the figures.

---

## [Editor Report · Decision Letter 2]

5 Oct 2022

Effectiveness of Two Extended-Release Buprenorphine Formulations during Postoperative Period in Neonatal Rats

PONE-D-22-06559R2

Dear Dr. Zhang,

We’re pleased to inform you that your manuscript has been judged scientifically suitable for publication and will be formally accepted for publication once it meets all outstanding technical requirements.

Kind regards,

Sairah Hafeez Kamran, PhD

Academic Editor

PLOS ONE
---

## [Editor Report · Acceptance letter]

6 Oct 2022

PONE-D-22-06559R2 

Effectiveness of Two Extended-Release Buprenorphine Formulations during Postoperative Period in Neonatal Rats 

Dear Dr. Zhang:

I'm pleased to inform you that your manuscript has been deemed suitable for publication in PLOS ONE. Congratulations! Your manuscript is now with our production department. 

Kind regards, 

on behalf of

Dr. Sairah Hafeez Kamran 

Academic Editor

PLOS ONE